# Hallucinations after Cardiac Surgery: A Prospective Observational Study

**DOI:** 10.3390/medicina56030104

**Published:** 2020-03-02

**Authors:** Thomas H. Ottens, Iris E.C. Sommer, Marieke J. Begemann, Maya Schutte, Maarten Jan Cramer, Willem J. Suyker, Diederik Van Dijk, Arjen J.C. Slooter

**Affiliations:** 1Department of Intensive Care Medicine, Haga Teaching Hospital, 40551 The Hague, The Netherlands; 2Department of Medical Sciences, Cells and Systems, University Medical Center Groningen, 9713 Groningen, The NetherlandsM.J.H.Begemann@umcutrecht.nl (M.J.B.); J.L.Schutte-2@umcutrecht.nl (M.S.); 3Department of Cardiology, University Medical Center Utrecht, 3584 Utrecht, The Netherlands; M.J.M.Cramer@umcutrecht.nl; 4Department of Cardiothoracic Surgery, University Medical Center Utrecht, 3584 Utrecht, The Netherlands; W.J.L.Suyker@umcutrecht.nl; 5Department of Intensive Care Medicine and UMC Utrecht Brain Center, University Medical Center Utrecht, 3584 Utrecht, The Netherlands; d.vandijk@umcutrecht.nl (D.V.D.);

**Keywords:** delirium, cardiothoracic surgery, hallucinations, neuropsychiatric outcomes

## Abstract

*Background and Objective:* Hallucinations after cardiac surgery can be a burden, but their prevalence and phenomenology have not been studied well. Risk factors for postoperative hallucinations, as well as their relation to delirium are unclear. We aimed to study the prevalence and phenomenology of hallucinations after cardiac surgery, and to study the association between hallucinations and delirium in this population. *Materials and Methods:* We used the Questionnaire for Psychotic Experiences to detect hallucinations in cardiac surgery patients and a control group of cardiology outpatients. We assessed postoperative delirium with validated instruments. Risk factors for postoperative hallucinations and the association between hallucinations and delirium were analysed using logistic regression. *Results:* We included 201 cardiac surgery patients and 99 cardiology outpatient controls. Forty-four cardiac surgery patients (21.9%) experienced postoperative hallucinations in the first four postoperative days. This was significantly higher compared to cardiology outpatient controls (*n* = 4, 4.1%, *p* < 0.001). Visual hallucinations were the most common type of hallucinations in cardiac surgery patients, and less common in outpatient controls. Cardiac surgery patients who experienced hallucinations were more likely to also have delirium (10/44, 22.7%) compared to patients without postoperative hallucinations (16/157, 10.2% *p* = 0.03). However, the majority of patients with postoperative hallucinations (34/44, 77.3%) did not develop delirium. *Conclusion:* After cardiac surgery, hallucinations occurred more frequently than in outpatient controls. Hallucinations after cardiac surgery were most often visual in character. Although postoperative hallucinations were associated with delirium, most patients with hallucinations did not develop delirium.

## 1. Introduction

Cardiac surgery often improves patients’ quality of life [1], but can be complicated by cerebral complications, including hallucinations [2,3,4]. Very little is known about the prevalence, risk factors and phenomenology of postoperative hallucinations. Clinical studies on hallucinations in cardiac surgery patients are sparse; we could only identify two case reports, one case series of three patients and one prospective observational study that included 52 patients [2,3,5,6]. The latter reported a high incidence of hallucinations and illusions of 58%, but the authors did not address any specific characteristics, such as modality or duration.

Hallucinations are sensory perceptions that occur in the absence of external stimulation of the relevant sensory organ, whilst a person is awake [7]. Hallucinations are seen in a variety of diseases, such as schizophrenia, blindness, Parkinson’s disease, and in up to 50% of patients with delirium [5,8]. Hallucinations can be frightening and may cause emotional distress [9].

Because hallucinations are a relatively unknown entity to clinicians, their presence may remain undetected or may be incorrectly attributed to delirium. Recently, the development of the Questionnaire for Psychotic Experiences (QPE), a validated instrument for the detection of hallucinations across all diagnostic groups, has enabled clinicians and researchers to screen for perioperative hallucinations in a structured fashion [10,11].

We have conducted a prospective observational study on the prevalence and phenomenology of hallucinations after cardiac surgery, using the QPE instrument. We also studied the relation between postoperative hallucinations and delirium, and compared the incidence of hallucinations with a non-surgical control group and investigated potential risk factors for postoperative hallucinations. We hypothesised that cardiac surgery patients would have a higher incidence of hallucinations than cardiology outpatient controls.

## 2. Materials and Methods

### 2.1. Study Design and Population

This single-center prospective observational study was carried out at the University Medical Center Utrecht, the Netherlands. The study protocol has been approved by the Institutional Review Board of the University Medical Center Utrecht (protocol number 16-602). This study was part of the Understanding Hallucinations project, which was funded by The Netherlands Organisation for Health Research and Development (ZonMW).

We recruited adult patients of 18 years or older who were scheduled for elective or semi-elective cardiac surgery at the pre-operative screening clinic. Exclusion criteria were inability to speak and understand Dutch or English, mental incompetence to give informed consent and a history of alcohol abuse. We also recruited a non-surgical control group. To improve comparability between surgical patients and controls on possible predisposing factors for postoperative hallucinations, controls were selected from the cardiology outpatient clinic. Patients under outpatient follow-up for coronary artery disease, valvular heart disease or cardiac arrhythmias were invited to participate. Exclusion criteria in this control group were the same as in the cardiac surgery group described above. Controls were also excluded if they had undergone cardiac surgery within the last 12 months. Patients in both groups provided written informed consent prior to participation.

### 2.2. Anaesthesia and Intensive Care in the Cardiac Surgery Group

Participants in the cardiac surgery group received care as per routine clinical practice. This included premedication with oral midazolam on the day of surgery, unless contra-indicated because of advanced age (>75 years), obstructive sleep apnea or a neuromuscular disorder. Anaesthesia induction and maintenance techniques were left at the discretion of the attending anaesthesiologist. As per local protocol, general anaesthesia was induced with sufentanil, midazolam and rocuronium and maintained with sufentanil and either isoflurane or sevoflurane. Postoperatively, all patients in this group were admitted to the intensive care unit (ICU).

### 2.3. Assessment of Hallucinations and Delirium

Cardiac surgery patients were followed up from the first until the fourth postoperative day. On each of these days, trained research assistants administered the QPE instrument. The validated hallucination screening instrument is described in detail elsewhere [10,11]. On the first postoperative day, we administered a short screening version of the QPE. If a patient’s response to any of the QPE screening questions suggested possible hallucinations, the full version of the QPE was administered on that day, as well as on all the subsequent days of the follow-up period. If the patient’s response to the QPE screening version on the first postoperative day did not suggest hallucinations, we administered the screening version again on the next postoperative day.

The occurrence of delirium after cardiac surgery was assessed by the research assistant who also administered the QPE. During their ICU stay, a research assistant screened patients twice daily using the Confusion Assessment Method for the ICU (CAM-ICU) and the Intensive Care Delirium Screening Checklist (ICDSC) [12,13]. The bedside nurse also screened patients for delirium using the CAM-ICU twice daily. Patients were not assessed for delirium if they were unarousable, as measured with the Richmond Agitation Sedation Scale (score –4 or –5) [14]. After discharge from the ICU, the bedside nurse screened patients for delirium once every shift using the Delirium Observation Scale (DOS) [15], and research assistants screened the patients using the CAM, once daily. Research assistants also reviewed the patient’s medical charts of the same observation period for notes that suggested the occurrence of delirium or administration of antipsychotic medication. The diagnosis of delirium was based on all available information, according to the diagnostic criteria described in the Diagnostic and Statistical Manual, edition IV-R (DSM-IV-R). The diagnosis was adjudicated by the study coordinator (TO) [7]. In case of any doubts, we consulted a neurologist-intensivist (AJS), who had the final vote.

A research assistant interviewed the participants in the control group in a private room. First, the screening version of the QPE instrument was administered. Similar to the cardiac surgery group, we administered the full version of the QPE when a participant’s response to the QPE screening questions suggested possible hallucinations. Control participants were asked to describe experiences with hallucinations in the past week. Control participants were not followed up after their interview.

All positive responses to the QPE were recorded in detail and discussed in a meeting with expert psychiatrists and researchers, before classifying the response as a hallucination. An experience was classified as a hallucination only when experienced in the awake state, with eyes open.

### 2.4. Assessment of Risk Factors

In all cardiac surgery patients, we recorded age, gender, type of cardiac disease, cardiac symptom severity (New York Heart Association (NYHA) class of disability from dyspnea or Canadian Cardiology Society (CCS) class of disability from chest pain), left ventricular ejection fraction, kidney function, as well as the presence of insulin-dependent diabetes, chronic pulmonary disease, extracardiac arterial occlusive disease and previous cardiac surgery. In the perioperative phase, we recorded the use of benzodiazepine premedication, the duration of cardiopulmonary bypass, occurrence of delirium (as described above), and transfusions during surgery or the postoperative observation period. We recorded the cumulative dose of all benzodiazepines during the postoperative phase, starting from the arrival of the patient in the ICU until the fourth postoperative day and converted the dose to midazolam equivalents [16]. Similarly, we added and converted all opioid doses during the postoperative phase and converted this to intravenous morphine equivalents [17].

### 2.5. Data Analysis

We used the Chi-square or Fisher’s exact test to compare the occurrence of hallucinations between the cardiac surgery group and the control group, where appropriate. To identify potential risk factors for postoperative hallucinations, we conducted bivariate analyses of demographic data, relevant comorbidities and cardiac disease characteristics and postoperative clinical variables. Frequencies were compared using the Chi-square test or Fisher’s exact test, means of normally distributed data with the student’s t-test, and medians of non-normally distributed data with the Mann-Whitney-U test. Variables with a univariable *p*-value of <0.10 were selected as potential risk factors. We evaluated the strength of the association of the risk factors after correction for each other’s influence by estimating their adjusted odds ratio’s with use of a multivariable logistic regression model. We used the Statistical Package for the Social Sciences (SPSS, IBM SPSS Inc., Chicago, IL, USA) version 22 for all analyses.

## 3. Results

### 3.1. Study Population

Between July 2013 and January 2015, 299 cardiac surgery patients gave informed consent for participation. 36 patients did not participate in the study because an exclusion criterion was discovered after the informed consent procedure, the surgery was cancelled, or the patient withdrew their consent. Logistic problems, such as early postoperative transferal to another hospital precluded follow-up in 51 patients. 8 patients could not be interviewed because of postoperative complications. Of these 8 patients, 5 could not be interviewed because their level consciousness (as measured by the RASS-score) was too low to allow hallucination or delirium observation. Of 3 patients, the case record form was incomplete. Therefore, 201 cardiac surgery patients were included in the analysis.

Between September 2015 and April 2016, 101 patients were recruited for participation in the cardiology outpatient group. Two of these patients were excluded from the analysis. One patient did not fulfil the inclusion criteria, another patient refused to complete the questionnaires. The baseline characteristics of cardiac surgery and cardiology outpatient group participants are shown in Table 1.

Table 2 shows the characteristics of the surgical procedures in the cardiac surgery group. During the first four postoperative days, 135 (67.2%) patients received a benzodiazepine at least once. The median cumulative amount of opioids administered during the observation period was 13.5 mg (interquartile range 5.0 to 28.0 mg) IV morphine equivalents. Most patients received morphine in the ICU, and oxycodone on the ward. 

### 3.2. Prevalence of Hallucinations

Of the 201 cardiac surgery patients, 44 (21.9%) experienced hallucinations during the postoperative follow-up period. By contrast, four participants (4.1%) in the cardiology outpatient group reported to have experienced hallucinations in the week prior to the interview. This difference was statistically significant (*p* < 0.01).

### 3.3. Phenomenology of Hallucinations

Of the 44 patients in the cardiac surgery group who experienced postoperative hallucinations, 31 patients (70.5%) reported isolated visual hallucinations. Other types of hallucinations reported postoperatively were auditory (*n* = 9, 20.5%), olfactory (*n* = 5, 11.4%) and tactile (*n* = 6, 13.6%) hallucinations. Seven patients (15.9%) experienced hallucinations of combined modalities. The reported hallucinatory experiences ranged from flowers that appeared on the walls, water flowing into the room and the appearance of a patient’s deceased spouse. One patient heard marching soldiers and people speaking in a different language, another patient heard children playing.

In contrast with the cardiac surgery patients, none of the cardiology outpatient group participants reported recent visual hallucinations. The hallucinations reported in this group were either isolated auditory (*n* = 2, 50%) or auditory combined with another sensory modality (*n* = 2, 50%).

When asked about the severity of the hallucinations, eight cardiac surgery patients (4.0%) reported that their visual or auditory hallucinations caused them at least some degree of discomfort or anxiety. Details about the duration, time of occurrence and complexity of the visual and auditory hallucinations in cardiac surgery patients are shown in Appendix A.

### 3.4. Hallucinations and Delirium in the Cardiac Surgery Group

Postoperative delirium was detected in 26 cardiac surgery patients (12.9%), as shown in Table 3. Patients who reported to have experienced postoperative hallucinations were more likely to have had at least one episode of delirium compared to those without hallucinations (22.7% vs. 10.2%, crude odds ratio (OR) 2.59, 95% confidence interval (95% CI) 1.08 to 6.21, *p* = 0.03). However, 34 of the 44 patients (77.3%) with postoperative hallucinations did not develop postoperative delirium.

### 3.5. Risk Factors for Postoperative Hallucinations

Univariate logistic regression analysis showed that postoperative hallucinations were associated with age (crude OR 1.05 per year, 95% CI 1.01 to 1.09, *p* = 0.01), duration of cardiopulmonary bypass (crude OR 1.005 per minute, 95% CI 1.00 to 1.01, *p* = 0.075), transfusion (crude OR 2.08, 95% CI 1.04 to 4.16). Patients who received benzodiazepine premedication had a lower risk of postoperative hallucinations (crude OR 0.32, 95% CI 0.15 to 0.68, *p* = 0.003). The effect of benzodiazepine premedication depended on age (logistic interaction *p* = 0.02). This may be explained by the local premedication protocol which does not recommend benzodiazepine premedication for patients older than 75 years.

The type of cardiac disease (coronary disease, valvular disease or arrhythmia), left ventricular ejection fraction, history of previous cardiac surgery, kidney function and the presence of extracardiac arteriopathy, insulin-dependent diabetes or chronic pulmonary disease were not associated with postoperative hallucinations. The cumulative postoperative opioid and benzodiazepine doses were also not significantly associated with postoperative hallucinations. Detailed descriptions of the univariate logistic regression analyses are presented in Appendix A.

A multivariable logistic regression model was fitted using the variables with significant associations to the occurrence of postoperative hallucinations mentioned above, as well as an interaction term for age and benzodiazepine premedication. The results of the analysis are presented in Table 4. None of the risk factors had significant associations with postoperative hallucinations after multivariable adjustment.

## 4. Discussion

In this prospective study of the prevalence, phenomenology and risk factors of postoperative hallucinations after cardiac surgery, we found that 21.9% of cardiac surgery patients experienced postoperative hallucinations. This was significantly higher than in cardiology outpatients (4.1%) with similar cardiovascular diseases and risk factors. The hallucinations reported by cardiac surgery patient were predominantly visual in character. Relatively few patients reported significant discomfort or anxiety as a result of the hallucinations. Although this study did not investigate the pathophysiology behind the occurence of postoperative hallucinations, the high incidence of hallucinations after cardiac surgery may be explained by many factors, such as the use of high doses of opioids during anesthesia, the administration of other hallucinogenic medications in the perioperative phase, and potentially also the systemic inflammation induced by the major surgery and cardiopulmonary bypass. More research, particularly in other populations, is needed to answer the question if the occurrence of postoperative hallucinations differs between patients who undergo cardiac surgery and other types of surgery.

The prevalence of delirium in our study was relatively low compared to older studies that reported delirium in up to 50% of cardiac surgery patients [18,19], but corresponded well to findings from other recent large studies in the cardiac surgery population [20,21,22]. Most patients with postoperative hallucinations did not have concomitant delirium, but the risk of delirium was higher in patients with hallucination compared to those without hallucinations. Univariable analysis also identified associations between postoperative hallucinations and older age, duration of cardiopulmonary bypass and postoperative administration of blood products, and a mild protective effect of benzodiazepine premedication. After multivariable adjustment for these factors, the association between postoperative hallucinations and delirium was no longer significant.

The administration of benzodiazepine premedication seemed to protect against postoperative hallucinations. This effect may be explained by the policy that such premedication was not given to older patients or those with particular comorbidities. Withholding of benzodiazepine premedication may thus be interpreted as a marker for age and comorbidity, rather than having an actual protective effect for hallucinations.

Strengths of this study are the large sample size and combined use of multiple validated diagnostic instruments to maximise diagnostic acuity for delirium. Repeating the QPE instrument daily minimised the risk of missing hallucinatory episodes due to the patient forgetting them along the postoperative phase, and helped to capture a maximum amount of detail about the hallucinatory experiences. Still, we believe this study could have underestimated the prevalence of hallucinations. Patients may be reluctant to admit having experienced hallucinations, because they may have feared stigmatisation. Patients seemed to experience little distress from their hallucinatory experiences in this study. In part, this may reflect underreporting from patients who were frightened by their hallucinations and therefore choose not to discuss them and report them during their observation sessions. Also, the assessment of neuropsychiatric outcomes in the postoperative phase can be hindered by complications such as re-operation or medical treatments that preclude testing. Postoperative cognitive dysfunction, delirium or the administration of memory impairing medication such as benzodiazepines may also have caused patients not to remember and subsequently report their hallucinatory experiences [23]. This is also relevant for the association we found between hallucinations and delirium, which should therefore be interpreted with caution.

This study had several limitations. For practical reasons, follow-up in the cardiac surgery group had to be limited to four days, whereas participants in the control group were asked to report hallucinations in the preceding week. Because the control group is not treated with surgery and not admitted to the hospital, the comparison with the cardiac surgery population includes the effects of hospitalisation and surgical trauma in general. This limits the generalisability of our findings. Also, a relatively high proportion of patients had to be excluded because their follow-up was incomplete. This may have led to under- or overestimation of the prevalence of hallucinations. However, the difference between the prevalence of recent hallucinations in the surgical group versus the control group was large enough to conclude that the high prevalence of hallucinations in the postoperative phase was actually related to the surgical procedure and perioperative factors including hospitalisation with ICU treatment. This conclusion is further supported by data from other projects within the Understanding Hallucinations project. For other purposes than this present study, the project recruited a control group of 500 healthy community dwelling persons (average age 45 years SD 20.3 years) the prevalence of recent hallucinations was 46 (9.2%), which is significantly lower than that in the cardiac surgery patients (*p* < 0.01) but not statistically significantly different than the prevalence found in the outpatient control group (*p* = 0.11). These differences will also in part relate to different testing circumstances, age and presence of unobserved risk factors. In particular our elderly cardiology outpatients may have forgotten hallucinations experienced when they were younger or may be embarrassed to report them. Because the study focused on a cardiac surgery populations, the analysis of risk factors focused on typical cardiac disease risk factors and peroperative factors related to this type of surgery. Other risk factors may still be present but could not be evaluated in this study.

## 5. Conclusions

In conclusion, we found a high prevalence of mostly visual hallucinations after cardiac surgery. Although there was an association between postoperative hallucinations and delirium, most patients with hallucinations did not develop postoperative delirium. In most cases, hallucinations did not cause severe distress or anxiety. Although there was an effect of age, duration of CPB, postoperative transfusion and benzodiazepine premedication in univariable regression analysis, none of these factors were independent predictors of postoperative hallucinations in a multivariable logistic regression model.

## Figures and Tables

**Table 1 medicina-56-00104-t001:** Demographic characteristics and baseline parameters of the participants in the cardiac surgery and cardiology outpatient groups.

	Cardiac Surgery Group *N* = 201	Cardiology Outpatient Group *N* = 99	*p*-Value
Demographics			
Age, years, mean (SD)	64.6 (11.0)	65.2 (12.2)	0.67
Male sex	146 (72.6)	64 (64.6)	0.16
Type of heart disease			
Isolated coronary artery disease	76 (37.8)	30 (30.3)	
Isolated heart valve disease	47 (23.4)	12 (12.1)	
Isolated cardiac arrhythmia	0 (0)	28 (28.3)	
Multiple cardiac pathologies	78 (38.8)	29 (29.3)	<0.001
Disease severity parameters			
History of cardiac surgery	13 (6.5)	35 (35.4)	<0.001
Symptom severity NYHA/CCS-class III or IV	69 (34.3)	27 (27.3)	0.22
Left Ventricular Ejection Fraction			
Good (>50%)	125 (62.2)	74 (74.4)	
Moderate (31–50%)	58 (28.9)	22 (22.2)	
Poor (<30%)	18 (9.0)	4 (4.0)	0.02
Comorbidities			
Impaired kidney function (eGFR<51 mL/min/1.73m2)	15 (7.5)	5 (5.1)	0.46
Extracardiac arteriopathy	18 (9.0)	16 (16.2)	0.06
Chronic use of inhaled bronchodilators or steroids for lung disease	19 (9.5)	8 (8.1)	0.70
Diabetes managed with insulin	16 (8.0)	1 (1.0)	0.03

All data presented as *N* (%), unless indicated otherwise. Abbreviations: CCS, Canadian Cardiovascular Society grading of angina pectoris. NYHA, New York Heart Association functional classification of heart failure symptoms. eGFR, estimated glomerular filtration ratio.

**Table 2 medicina-56-00104-t002:** Surgical characteristics of the cardiac surgery group (*n* = 201).

Type of Surgery	
Coronary Artery Bypass Grafting, *N* (%)	88 (43.8)
Single Valve, *N* (%)	48 (23.9)
Multiple Valves, *N* (%)	7 (3.5)
Thoracic Aorta, *N* (%)	8 (4.0)
Combined procedures, *N* (%)	50 (24.9)
Duration of cardiopulmonary bypass, median (IQR), minutes	105 (80 to 147)
EuroSCORE-II (predicted mortality), median (IQR), %	1.5 (0.9 to 2.9)
Postoperative transfusion of blood products, *N* (%)	61 (30.3)
Received benzodiazepine premedication, *N* (%)	164 (81.6)
Cumulative postoperative benzodiazepine dose (midazolam equivalents),mg	5 (0 to 10)
Cumulative postoperative opioid dose (morphine equivalents), mg	13.5 (5 to 28)

Cumulative drug doses calculated from postoperative arrival in the intensive care unit until the end of the follow-up period.

**Table 3 medicina-56-00104-t003:** Occurrence of hallucinations and delirium in the cardiac surgery group.

	Delirium Present	Delirium Absent	Total
Hallucinations present	10	34	44
Hallucinations absent	16	144	160
Total	26		204

**Table 4 medicina-56-00104-t004:** Logistic regression analysis for identification of risk factors of postoperative hallucinations in the cardiac surgery group (*N* = 201) ^a^.

	Patients without Hallucinations *N* = 157	Patients with Hallucinations *N* = 44	Wald	OR_adj_	95% CI	*p*-Value
Age, years, median (IQR)	64 (57 to 72)	69 (62 to 78)	0.47	1.03	0.94 to 1.07	0.50
Benzodiazepine premedication	135 (86)	29 (65.9)	0.01	0.70	0.13 to 0.93	0.91
Duration of cardiopulmonary bypass, minutes, mean (SD)	114 (54)	133 (78)	1.21	1.003	1.00 to 1.01	0.27
Postoperative transfusion of blood products	42 (26.8)	19 (43.2)	1.16	1.54	0.70 to 3.36	0.28
Postoperative delirium	16 (10.2)	10 (22.7)	0.49	1.42	0.53 to 3.81	0.48

^a^ The dependent variable in the model is postoperative hallucinations (yes/no). All values are presented as number (%), unless otherwise indicated. Overall model Chi-square 15.85, *p* = 0.015. Nagelkerke R-square 0.12, Hosmer and Lemeshow chi-square for goodness of fit 5.77, *p* = 0.67. Interaction term for age and benzodiazepine premedication *p* = 0.87. Abbreviations: IQR; inter-quartile range, ORadj, adjusted odds ratio; SD, standard deviation; 95%CI, 95% confidence interval.

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
