# Peer review of "Hallucinations after Cardiac Surgery: A Prospective Observational Study"

_medicina, 2020, doi:10.3390/medicina56030104_

Round 1

Reviewer 1 Report

This is an important prospective observational study examining the prevalence of hallucinations after cardiac surgery and their association with postoperative delirium. This is a well-written and described manuscript and is important for delirium science. I have few comments for the authors. 

What was the mean or IQR of RASS scores for the ICU population? I am curious if those that had hypoactive delirium, or negative RASS scores, were unable to fully verbalize their hallucinations due to their level of arousal. This might be an important consideration for a descriptive table and a limitation.  Was diluadid or hydromorphone administration information collected/included in the overall opioid dose category? Are the authors able to expand on what opioids were administered? Also, was ketamine administration collected? Both of these analgesics have noted side effects of hallucinations.  I did not see this mentioned in the manuscript; was psychiatric history collected?  Page 8, Line 203-crude OR is missing p value for transfusion

Author Response

Dear sir / madam,

Thank you very much for taking the time to review our manuscript, titled "Hallucinations after cardiac surgery: a prospective observational study". We appreciate your comments and will respond to them as best we can. 

  1. We agree that the RASS-score would give an interesting insight in the participants' ability to cooperate with the study procedures. However, we only registered if RASS-scores were too low to perform delirium screening and/or hallucination interviews. Specifically, 5 patients had RASS-scores that were consistently too low to screen for delirium or hallucinations during their 4 day postoperative observation period. For clarity, we have added this information to the results section.
  2. Hydromorphone (Dutch trade name Palladon) is not available for postoperative pain relief at the hospital where this study was carried out. Analgesics are administered according to strict ICU and cardiothoracic ward policies that allow for IV morphine in ICU patients and oral oxycodone in ward patients. Only in sporadic cases, piritramide (Dipidolor) was used because of morphine intolerance/allergy. 
  3. Ketamine is not available for medical use in The Netherlands. S-ketamine (Ketanest) was not used for pain relief after cardiothoracic surgery at the time this study was carried out.
  4. A detailed pyschiatric history was not carried out, because we expected difficulties with ethical approval for this otherwise relatively simple observational study. Recording a detailed psychiatric history in surgical patients is considered a serious burden and may not have been considered appropriate for privacy reasons. We therefore only inquired after alcohol intake and excluded patients if they abused alcohol.
  5. There must have been an error in the file, we have checked but could not find the omission. The manuscript text at 3.5 states the crude OR for transfusion correctly. Table 4 states the adjusted OR for transfusion correctly. 

Once again, thank you for your valuable comments. We have adjusted the manuscript accordingly. 

On behalf of the study authors,

Dr. Thomas Ottens

Reviewer 2 Report

I am interested in evaluating hallucinations after cardiac surgery using QPE. This study is an interesting topic. However, the research design of this study has an error. This study is the subject of hallucinations after cardiac surgery. I do not think it is right for the Outpatient group to be included in the study. Because the Outpatient and Surgery groups are subjects with different factors. Comparing these two groups has nothing to do with this study topic. If you want to be reviewed for this study, I will modify the subjects and rerun the statistics. And I hope to rewrite the research results.   I expect your findings to be this sort of distinction.  First, you should analyze the subject only with the Cardiac Surgery group (n = 201). Please present the demographic characteristics (including Surgical characteristics) of the Cardiac Surgery group and the characteristics according to the presence or absence of hallucinat. Supplemental Material table 1 is an important result. Please present this in the text. And table 4 should be presented. Please revise the entire text again to reflect the revised research results.  
Your topic is very interesting and necessary.
However, the study design includes unnecessary subjects, and the findings are difficult to read, making it difficult to understand the point of research.  There are errors in the method and results, but fortunately this study is expected to have good results if you reorganize the results

Author Response

The issues regarding this reviewer's comments and demands have been discussed via e-mail with the handling editor, Mr Matic.

Reviewer 2 proposes to modify our dataset, rerun the analysis and rewrite our paper him/herself. Although we are grateful for any constructive criticism and are willing to adjust our manuscript accordingly, we are sorry that we cannot comply with these requests.

We respectfully disagree with the reviewer that the control group should not have been included in this study. To identify a risk factor, it is necessary to include controls. In our study, inclusion of a control group allowed us to demonstrate that not just the cardiovascular disease of our patients, but surgery, anaesthesia and postoperative treatments induced hallucinations in almost 25% of patients. Please note that reviewer 1 and 3 have no comments on our study design choices regarding the choice of control participants.

Further, we suggest not to burden readers of this paper with the inclusion of all data of Supplemental table 1, as this is very large (over 2 full A4 pages) and detailed. In addition, we disagree with reviewer 2 that there is a necessity to present tables 1 and 2 separately for cardiac surgery patients with and without hallucinations, because we already present univariable and multivariable analyses on the associations of all these risk factors and the occurrence of hallucinations. In our opinion, splitting out the tables takes away to possibility to compare the patients with the controls and may confuse readers about potential effects of certain demographic factors, where in fact our analyses do not show an effect.

Reviewer 3 Report

I would like to thank the Editor for the opportunity to review the manuscript titled "Hallucinations after cardiac surgery: a prospective observational study" for publication in Medicina.

Overall it is a very well written manuscript that approaches a previously known but poorly described problem with rigor implementing a novel screening tool for hallucinations in a high risk population. The authors do an outstanding job in trying to account for potential confounders and sufficient explication for the associations found in their study and they should be commended for their work.

There are a few minor comments I would like to make:

Although this is an observational study and by no means can it be used to produce mechanistic evidence, it would be interesting if the authors could provide their insights on the pathophysiology of hallucinations after cardiac surgery (eg. neuroinflammation after cardiopulmonary bypass, etc.). The authors describe hallucinations as potentially frightening in the Introduction yet close to 77% of the patients in the study with visual hallucinations and 89% with auditory hallucinations describe them as not disturbing. This, as the authors hint in the discussion, might be explained by the fact that those patients with distressing hallucinations are not willing to talk about them or acknowledge them. It would be interesting for the reader to have the authors' comments on this issue. The reported incidence of hallucinations in the control individuals recruited for the Understanding Hallucinations is project close to 9% according to the authors and it is only 4% among the controls for the present study. Although this is not statistically significant it does seem like a remarkable difference and the article could benefit of an expanded explanation for this. 

Author Response

Thank you very much for the positive remarks and constructive criticisms.

  • We have certain theories - but not proof - on why cardiac surgery patients experience hallucinations so frequently. We have added a comment on our theory in the discussion section of the paper. In short, be believe that there may be influences from
    1. the administration of high doses of opioids during anesthesia for this type of surgery (a balanced anesthetic based on high doses of opioids)
    2. the impact of major surgery and cardiopulmonary bypass, which both induce systemic inflammation
    3. the administration of other hallucinogenic medication, such as anticholinergics. 
  • Thank you for your suggestion on the potential effect of frightening hallucinations not being reported by the participant. We have added a note on this factor in the discussion section.
  • In our opinion, the age of the control group mostly explains why this particular population seems to have a lower frequency of hallucinations compared to the healthy controls in the UH project. Because most people's first non-hospital admission related hallucinations occur at a relatively young age, some people in our control group may not remember or may be embarrassed to report hallucinations from their youth. We have added a note on this in our discussion section.